# Self-Supervised Graph Learning via Spectral Bootstrapping and Laplacian-Based Augmentations

## Abstract

We present LaplaceGNN, a novel self-supervised graph learning framework that bypasses the need for negative sampling by leveraging spectral bootstrapping techniques. Our method integrates Laplacian-based signals into the learning process, allowing the model to effectively capture rich structural representations without relying on contrastive objectives or handcrafted augmentations. By focusing on positive alignment, LaplaceGNN achieves linear scaling while offering a simpler, more efficient, self-supervised alternative for graph neural networks, applicable across diverse domains. Our contributions are twofold: we precompute spectral augmentations through max-min centrality-guided optimization, enabling rich structural supervision without relying on handcrafted augmentations, then we integrate an adversarial bootstrapped training scheme that further strengthens feature learning and robustness. Our extensive experiments on different benchmark datasets show that LaplaceGNN achieves superior performance compared to state-of-the-art self-supervised graph methods, offering a promising direction for efficiently learning expressive graph representations.

## 1 Introduction

Graph neural networks (GNNs) have emerged as powerful tools for learning representations of graph-structured data, with applications ranging from molecular property prediction to social network analysis. Although supervised learning approaches have shown impressive results, they often require large amounts of labeled data, which can be expensive or impractical to obtain in many real-world scenarios. This has led to increased interest in self-supervised learning methods for graphs, which aim to learn meaningful representations without requiring explicit labels. Recent advances in self-supervised graph learning have primarily focused on contrastive learning approaches, where models learn by maximizing agreement between different views of the same graph while minimizing similarity with negative samples. However, these methods face several key challenges. First, they often rely on handcrafted augmentation schemes that require careful tuning and domain expertise, making them difficult to adapt across different types of graphs and tasks. Second, the need for large negative sample sets introduces quadratic computational complexity with respect to graph size, limiting scalability to large-scale applications. Moreover, most existing approaches lack robustness against adversarial perturbations, making them vulnerable to structural attacks that can significantly degrade performance. Adversarial attacks on molecular graphs are particularly meaningful in real-world applications such as drug discovery and material design, where robustness is critical for safety and reliability. For instance, subtle adversarial modifications to molecular structures can mislead property prediction models (e.g., toxicity or efficacy) into producing erroneous results, which could have serious consequences in pharmaceutical development (Dai et al., 2018). Studying these vulnerabilities helps uncover blind spots in GNNs and motivates the development of more reliable models for high-stakes domains (Zügner et al., 2018). Additionally, adversarial attacks can serve as a tool to evaluate and improve model generalizability, ensuring robustness against noisy or maliciously perturbed inputs in real-world deployment (Sun et al., 2022; Guerranti et al., 2023).

To address these limitations, we introduce LaplaceGNN, a novel self-supervised framework that combines spectral bootstrapping with adversarial training to learn robust graph representations while maintaining linear computational complexity. Our approach makes three key contributions:

- We propose a principled method for generating graph views through Laplacian optimization and centrality-guided augmentations. This eliminates the need for handcrafted augmentations by automatically learning task-appropriate transformations based on spectral properties and node importance measures.

- We develop an adversarial teacher-student architecture that ensures robust feature learning through knowledge transfer. Unlike traditional contrastive approaches, our method leverages bootstrapped learning to avoid the computational burden of negative sampling while outperforming state-of-the-art performance.

- We rely on a bootstrapping scheme to keep $O(N)$ complexity by eliminating the need for explicit negative samples, making our approach scalable while preserving performance guarantees.

Extensive experiments across molecular, social, and biological networks demonstrate that LaplaceGNN achieves greater performance compared to existing methods. Our approach not only improves accuracy on standard benchmarks but also shows enhanced robustness against various types of adversarial attacks. Furthermore, transfer learning experiments reveal that the representations learned by LaplaceGNN are more generalizable, leading to significant improvements when pretrained on large-scale datasets and fine-tuned on downstream tasks.

The rest of the paper is organized as follows: Section 2 discusses related work on self-supervised graph learning and adversarial training. Section 3 presents the detailed methodology of LaplaceGNN, including the centrality-guided Laplacian optimization and the adversarial bootstrapping training components. Section 4 presents comprehensive experimental results across various datasets and tasks. Section 5 provides theoretical analysis and ablation studies. Finally, Section 6 concludes with a discussion of future research direction and improvements.

## 2 Related Work

Our work intersects with several key research directions in graph representation learning, which we review below.

**Self-supervised Learning on Graphs**  Self-supervised learning on graphs has emerged as a powerful paradigm for learning representations without relying on manual labels. Early approaches like DGI (Velickovic et al., 2019a) and InfoGraph (Sun et al., 2019) adapted mutual information maximization principles to the graph domain. Recent methods have explored contrastive learning, with works like GRACE (Zhu et al., 2020) and GraphCL (You et al., 2020) utilizing augmented graph views as positive pairs. BGRL (Thakoor et al., 2021) introduced a bootstrap approach to eliminate the need for negative samples, which we rely on in Section 5, while AD-GCL (Suresh et al., 2021a) incorporated adversarial training into contrastive learning. SSGE (Liu et al., 2025b) proposes a negative-free approach that achieves uniformity by aligning node representations with an isotropic Gaussian distribution. More recently, masked autoencoder approaches like GraphMAE (Hou et al., 2022) and AugMAE (Wang et al., 2024) have demonstrated strong performance by reconstructing masked graph components. More recently, masked autoencoder approaches like GraphMAE (Hou et al., 2022) and AugMAE (Wang et al., 2024) have demonstrated strong performance by reconstructing masked graph components.

**Graph Augmentation Strategies**  Graph augmentation techniques play a crucial role in self-supervised learning. Traditional approaches rely on handcrafted transformations such as edge dropping, feature masking, multi-views, and subgraph sampling (Zhu et al., 2021a; Hassani & Khasahmadi, 2020; Qiu et al., 2020). GCA (Zhu et al., 2021b) proposed adaptive augmentation strategies based on node centrality measures. Recent work has explored learnable augmentation policies, with methods like JOAO (You et al., 2021) and SP$^2$GCL

(Bo et al., 2024) automatically discovering optimal transformation combinations. These approaches, however, often require extensive tuning or may not generalize well across different graph types.

**Adversarial Training and Attacks on Graphs**   Adversarial training has emerged as a powerful technique to improve model robustness and generalization. In the graph domain, early work focused on defending against structural attacks (Zügner et al., 2018). Recent approaches like FLAG (Kong et al., 2020) and GRAND (Thorpe et al., 2022) have adapted adversarial training for representation learning. These methods typically generate perturbations at the feature or structure level, but often struggle with scalability on large graphs. InfMax (Ma et al., 2022) draws a connection between this type of attacks and an influence maximization problem graphs, proposing a group of practical attack strategies. Recent work has explored more efficient adversarial training schemes, such as virtual adversarial training (Zhuo et al., 2023) and diffusion-based approaches (Gosch et al., 2024a). In Section 4.4, we assess the robustness of our methods against adversarially poisoned input graphs, focusing on representations learned from graphs compromised by various structural attack strategies. These strategies encompass random attacks that randomly flip edges; DICE (Waniek et al., 2018), which deletes edges internally and connects nodes externally across classes; GFAttack (Chang et al., 2020), which maximizes a low-rank matrix approximation loss; and Mettack (Gosch et al., 2024b), which maximizes the training loss via meta-gradients.

**Spectral Graph Methods**   Spectral graph theory has long been fundamental to graph analysis and learning. Classical works established the connection between graph Laplacian eigenvalues and structural properties (Chung, 1997). In the context of GNNs, spectral approaches have been used for both architecture design (Defferrard et al., 2016) and theoretical analysis (Balcilar et al., 2021). Recent works have explored spectral properties for graph augmentation (Ghose et al., 2023) and robustness (Bo et al., 2024). However, the computational complexity of eigen-decomposition has hindered their application on large-scale graphs.

**Teacher-Student Framework**   Teacher-student architectures have shown promise in self-supervised learning, initially popularized in computer vision (Grill et al., 2020). In the graph domain, BGRL (Thakoor et al., 2021) adapted this approach to eliminate the need for negative sampling. Recent works have explored variations such as multi-teacher setups (Liu et al., 2025a) and cross-modal knowledge distillation (Dong et al., 2023). However, these approaches typically don't consider adversarial robustness or spectral properties in their design.

Our work bridges these research directions by introducing a novel framework that combines spectral graph theory with adversarial teacher-student training, while maintaining computational efficiency through bootstrap learning. Unlike previous approaches that rely on handcrafted augmentations or negative sampling, we propose a principled method for generating graph views through centrality-guided augmentations via Laplacian optimization.

# 3   Methodology of LaplaceGNN

We present LaplaceGNN, a self-supervised framework for graph representation learning that leverages spectral properties for robust feature learning. Given an input graph $\mathcal{G} = (V, E)$ with adjacency matrix $\mathbf{A} \in \{0, 1\}^{n \times n}$ and node features $\mathbf{X} \in \mathcal{R}^{n \times d}$, our goal is to learn representations that capture both structural and attribute information without relying on manual augmentations or negative samples. LaplaceGNN consists of three main components: (1) a spectral augmentation module that generates centrality-guided views to avoid tuning handcrafted transformations, (2) an adversarial online (teacher) network that ensures robust representations while the target (student) network learns through knowledge distillation, and (3) a linear bootstrapping method that eliminates the quadratic cost needed for negative sampling. Both Laplacian augmentations and adversarial bootstrapped learning schemes are described in Algorithm 1 and Algorithm 2, respectively.

### 3.1 Centrality-Based Augmentation Scheme

Unlike previous approaches that rely on fixed heuristic augmentations, we propose a principled method for perturbation based on centrality measures and spectral graph theory, allowing our framework to be adaptable to various graph structures and downstream tasks. Our key insight is that meaningful graph views should effectively maximize spectral differences while preserving critical structural properties by introducing controlled variations based on centrality-guided augmentations. We propose a novel centrality-guided augmentation scheme that combines multiple centrality measures to capture different aspects of node $i$ importance.

Let $C(i)$ represent the centrality score of node $i$, derived from a set of centrality functions $\mathcal{C} = \{C_1, C_2, \ldots, C_K\}$. Each $C_k$ corresponds to a centrality measure such as degree, PageRank, Katz centrality, or any other task-specific centrality measure relevant to the problem domain. The generalized centrality score is given by:

$$C(i) = \sum_{k=1}^{K} \alpha_k \cdot C_k(i), \tag{1}$$

where $\alpha_k$ are learnable weights that adjust the contribution of each centrality measure. The combined centrality matrix $\mathbf{C}$ is constructed from these scores, normalized to $[0, 1]$, and used to guide the augmentation process. This general formulation allows our method to flexibly adapt to diverse graph types and tasks. Importantly, the centrality-based approach ensures that modifications are focused on nodes critical for preserving the graph's structural and functional integrity, while creating challenging augmented views for robust learning. The goal is "controlled disruption": making meaningful changes to challenge the model while staying within bounds that preserve core graph properties. Spectral constraints ensure global characteristics are maintained, and focusing on important nodes generates stronger learning signals. This approach leads to robust node embeddings stable under augmentations, better generalization by handling structural variations, and stronger feature extractors capturing invariant properties.

Key advantages of this general approach include: flexibility to integrate domain-specific centrality measures tailored to downstream tasks, a principled mechanism for balancing global and local structural augmentations, and preservation of critical graph properties while introducing meaningful variations. The full rationale, along with the importance of the inclusion of centrality-based augmentation scheme, are given in Appendix A and Appendix A.1.1.

### 3.2 Laplacian Optimization

We formulate view generation as a Laplacian optimization problem that refines centrality-based probabilities (Algorithm 1). The optimization process begins by computing a combined centrality matrix $\mathbf{C}$ encoding structural node importance through multiple centrality metrics. This initialization provides a principled starting point for the augmentation matrices $\Delta_1$ and $\Delta_2$, where the outer product $\mathbf{C}\mathbf{C}^T$ prioritizes edges connecting high-centrality nodes.

The core optimization operates on the normalized graph Laplacian $\tilde{\mathbf{L}} = \mathbf{I} - \tilde{\mathbf{A}}$, derived from $\tilde{\mathbf{A}} = \mathbf{D}^{-1/2}\mathbf{A}\mathbf{D}^{-1/2}$. The goal is to compute two augmented Laplacians using the following sequential optimization strategy.

**Maximization View:** $\Delta_1$ maximize spectral divergence via gradient ascent:

$$\Delta_1^{(t+1)} \leftarrow \text{Proj}_{[0,1],B} \left( \Delta_1^{(t)} + \eta \nabla_{\Delta_1} \|\lambda(\tilde{\mathbf{L}}_{\text{mod1}})\|_2 \right),$$

where $\tilde{\mathbf{L}}_{\text{mod1}} = \tilde{\mathbf{L}} + \tilde{\mathbf{L}}(C\Delta_1 C^T)$.

**Minimization View:** $\Delta_2$ minimize divergence via gradient descent:

$$\Delta_2^{(t+1)} \leftarrow \text{Proj}_{[0,1],B} \left( \Delta_2^{(t)} - \eta \nabla_{\Delta_2} \|\lambda(\tilde{\mathbf{L}}_{\text{mod2}})\|_2 \right),$$

**Algorithm 1** Laplace Centrality Augmentations

1: **Input:** Normalized Laplacian matrix $\tilde{\mathbf{L}}_1$, budget constraint ratio $r$, learning rate $\eta$, iterations $T$.
2: **Parameters:** Set of centrality measures $\{C_k\}_{k=1}^K$, centrality weights $\{\alpha_k\}_{k=1}^K$.
3: **Compute budget:**
4: $B \leftarrow r \cdot (\sum_{i,j} \mathbf{A}_{ij})/2$
5: **Compute centrality:**
6: **for** each $C_k$ **do**
7:     Compute centrality scores (e.g., degree, PageRank, Katz)
8:     Combine centralities: $C(i) \leftarrow \sum_{k=1}^K \alpha_k C_k(i)$
9: **end for**
10: Normalize $\mathbf{C}$ to $[0, 1]$.
11: **Initialize Centrality-guided Augmentations:**
12: $\Delta_1 \leftarrow \mathbf{C}\mathbf{C}^T$ for lower triangular entries.
13: $\Delta_2 \leftarrow \mathbf{C}\mathbf{C}^T$ for lower triangular entries.
14: **Laplace Optimization:**
15: **for** $t = 1$ to $T$ **do**
16:     $\tilde{\mathbf{L}}_{\text{mod}1,2} \leftarrow \tilde{\mathbf{L}} + \tilde{\mathbf{L}}(C\Delta_{1,2}C^T)$
17:     Compute spectrum $\lambda_{mod1} = \lambda(\tilde{\mathbf{L}}_{\text{mod}1})$
18:     **Max:** $\Delta_1^t \leftarrow \text{Proj}_{[0,1],B}(\Delta_1^{t-1} + \eta\nabla_{\Delta_1}\|\lambda_{mod1}\|^2)$
19:     Compute spectrum $\lambda_{mod2} = \lambda(\tilde{\mathbf{L}}_{\text{mod}2})$
20:     **Min:** $\Delta_2^t \leftarrow \text{Proj}_{[0,1],B}(\Delta_2^{t-1} - \eta\nabla_{\Delta_2}\|\lambda_{mod2}\|^2)$
21: **end for**
22: **Generate Augmented Views:**
23: **for** each graph $G$ in dataset **do**
24:     Sample $P_1 \sim \mathcal{B}(\Delta_1)$, $P_2 \sim \mathcal{B}(\Delta_2)$
25:     Generate views: $\tilde{\mathbf{L}}_1 = \tilde{\mathbf{L}} + \tilde{\mathbf{L}}(CP_1C^T)$,
    $\tilde{\mathbf{L}}_2 = \tilde{\mathbf{L}} + \tilde{\mathbf{L}}(CP_2C^T)$
26: **end for**
27: **Output:** Augmented Laplacian matrices $\tilde{\mathbf{L}}_1$, $\tilde{\mathbf{L}}_2$.

**Algorithm 2** Adversarial Bootstrapped GNN

1: **Input:** Graph $\mathcal{G} = (\mathcal{V}, \mathcal{E})$, feature matrix $\mathbf{X}$, learning rate $\eta_\xi$, perturbation bound $\epsilon$, ascent steps $T$, step size $\alpha$, decay rate $\beta$.
2: **Initialize:** Teacher parameters $\xi$; EMA $\theta \leftarrow \xi$.
3: **for** each training epoch **do**
4:     Given two views $\hat{\mathcal{G}}_1, \hat{\mathcal{G}}_2$ via Algorithm 1:
5:     **Forward Pass (Teacher Encoder with MLP Projector):**
6:     $\hat{\mathbf{T}} = \text{MLP}_\xi(q_\xi(g_\xi(\hat{\mathcal{G}}_1, \mathbf{X})))$
7:     **Forward Pass (Student Encoder):**
8:     $\hat{\mathbf{Z}} = p_\theta(f_\theta(\hat{\mathcal{G}}_2, \mathbf{X}))$
9:     **Compute the bootstrapped loss:**

$$\mathcal{L}_{\text{boot}} = -\frac{2}{N}\sum_{i=0}^{N-1}\frac{\hat{\mathbf{T}}^{(i)}\hat{\mathbf{Z}}^{(i)\top}}{\|\hat{\mathbf{T}}^{(i)}\|\|\hat{\mathbf{Z}}^{(i)}\|}$$

10:     **Adversarial Training for Teacher Model:**
11:     Initialize perturbation $\boldsymbol{\delta}_0 \sim \mathcal{U}(-\epsilon, \epsilon)$.
12:     $\mathbf{H} = g_\xi(\hat{\mathcal{G}}_1, \mathbf{X})$ (hidden features of the teacher encoder)
13:     **for** $t = 1, \ldots, N_{\text{epoch}}/T$ **do**
14:         Compute:
        $\mathbf{g}_t \leftarrow \mathbf{g}_{t-1} + \frac{1}{T}\nabla_\xi\mathcal{L}_{\text{boot}}(\hat{\mathcal{G}}_1, \mathbf{H} + \boldsymbol{\delta}_{t-1})$
15:         Compute gradient:
        $\mathbf{g}_\delta = \nabla_\delta\mathcal{L}_{\text{boot}}(\hat{\mathcal{G}}_1, \mathbf{H} + \boldsymbol{\delta}_{t-1})$
16:         Update perturbation:

$$\boldsymbol{\delta}_t = \text{Proj}_\epsilon\left(\boldsymbol{\delta}_{t-1} + \alpha\frac{\mathbf{g}_\delta}{\|\mathbf{g}_\delta\|_F}\right)$$

17:     **end for**
18:     **Update Parameters:**
19:     Teacher Update: $\xi \leftarrow \xi - \eta_\xi\mathbf{g}_T$
20:     EMA Update: $\theta \leftarrow \beta\theta + (1 - \beta)\xi$
21: **end for**

with $\tilde{\mathbf{L}}_{\text{mod}2} = \tilde{\mathbf{L}} + \tilde{\mathbf{L}}(C\Delta_2C^T)$.

Then, the projection operator $\text{Proj}_{[0,1],B}$ enforces a budget $B = r \cdot (\sum_{i,j} \mathbf{A}_{ij})/2$, with $r$ being the budget ratio, to limit total augmentation, while $\mathbf{C}\Delta\mathbf{C}^T$ prioritizes high-centrality edges (full optimization steps in Appendix A.2). Then, the final augmented views are generated through Bernoulli sampling and the sampled $P_1, P_2$ are applied to create the augmented Laplacian matrices:

$$\tilde{\mathbf{L}}_{\text{mod}1} = \tilde{\mathbf{L}} + \tilde{\mathbf{L}}(C\Delta_1C^T), \quad \tilde{\mathbf{L}}_{\text{mod}2} = \tilde{\mathbf{L}} + \tilde{\mathbf{L}}(C\Delta_2C^T), \tag{2}$$

ensuring that the max-view disrupts spectral properties to challenge the model, while the min-view preserves core structure for stable learning. This dual strategy balances adversarial robustness with structural fidelity, as detailed in Appendix A.2.1.

### 3.3 Self-Supervised learning via Adversarial Bootstrapping

We propose an adversarial bootstrapping framework that combines teacher-student knowledge transfer with adversarial training to learn robust graph representations. Unlike traditional contrastive approaches that rely on negative sampling, our method leverages a bootstrapped learning scheme where the teacher network generates robust targets from which the student network can learn. Similar to other bootstrapping meth-

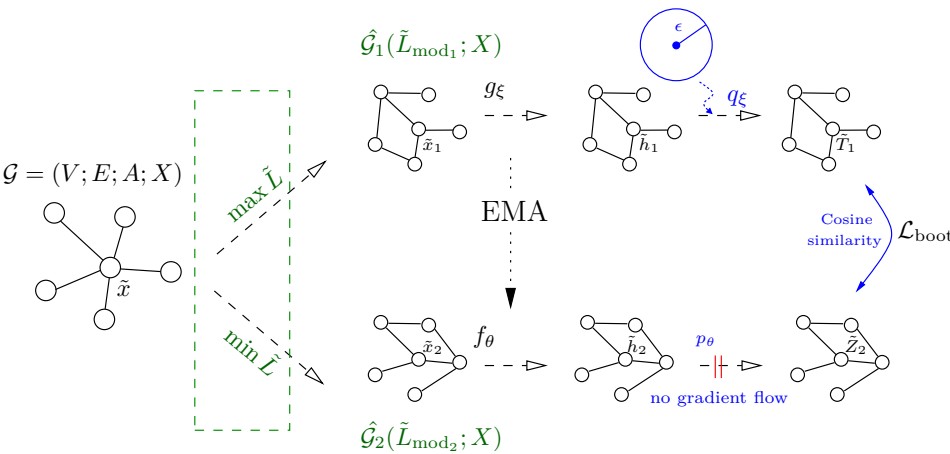

Figure 1: Illustration of the adversarial bootstrapping framework in LaplaceGNN. To enhance robustness, teacher encoder $g_\xi$ receives adversarial perturbations $\boldsymbol{\delta}_t$, bounded by $\epsilon$, while student encoder $f_\theta$ learns through knowledge transfer via EMA of teacher's weights. To avoid collapse, both encoders project their hidden representations through MLP heads $q_\xi$ and $p_\theta$, for teacher and student respectively, and their outputs are compared using cosine similarity $\mathcal{L}_{\text{boot}}$. The student's gradients flows are stopped during training, ensuring stable updates.

ods Thakoor et al. (2021), we eliminate the need for negative samples that typically introduce quadratic computational complexity in the size of the input.

However, we extend beyond this approach by combining their momentum encoder with an adversarially trained teacher network to introduce spectral constraints that ensure more meaningful positive pairs and maintain $O(N)$ memory complexity. The advantage of employing adversarial training is to provide stronger resistance to structural attacks, more stable gradients during training (due to the removal of negative sampling), and better generalization on transfer learning. On the one hand, our framework keeps the memory efficiency by eliminating the computational burden of negative sampling; on the other hand, it creates more challenging yet meaningful positive pairs via teacher adversarial perturbations. These improvements allow LaplaceGNN to scale linearly in the size of the input while maintaining robustness against adversarial attacks, addressing two key limitations in current graph representation learning approaches. The overall training scheme is shown in Algorithm 2.

### 3.3.1 Adversarial Bootstrapping

Our framework incorporates adversarial training within a self-supervised learning context to enhance robustness and generalization without relying on handcrafted views informed by domain expertise. Given an input graph $\mathcal{G} = (V, E)$ with adjacency matrix $\mathbf{A} \in \{0,1\}^{n \times n}$ and node features $\mathbf{X} \in \mathcal{R}^{n \times d}$, our objective is to learn an embedded representation $\mathbf{H} \in \mathbb{R}^{n \times d}$ to be used for downstream tasks, such as graph and node classification. For the sake of simplicity and notation coherence, we adopt the node classification formalism throughout this section. However, our approach has been straightforwardly generalized to graph classification setup, as shown in Section 4.

The teacher encoder $g_\xi$ and student encoder $f_\theta$ form a dual-encoder architecture, where the teacher (target) network and student (online) network share the same architectural blueprint but with the gradients flowing only through the teacher, as shown in Figure 1. Adversarial perturbations $\boldsymbol{\delta}_t^{\mathbf{H}}$ are applied to hidden representations, optimized via projected gradient descent. We refer to Appendix A.3.2 for details on adversarial bootstrapping deployment. The student network's parameters are updated through EMA of the teacher's parameters and followed by a projector $p_\theta$. To prevent representation collapse, the teacher network incorporates an additional MLP projector $q_\xi$, followed by a final MLP head. Therefore, LaplaceGNN's training procedure alternates between two phases, outer minimization through the bootstrapped loss $\mathcal{L}_{\text{boot}}$ and inner

Table 1: Node classification performance measured by accuracy%, with standard deviations over 10 random seed runs. The best and second-best results are shown in bold and underlined, respectively. For sake of space, this table has been shortened and the full version is available in Appendix C, Table 8.

| DATASET | WIKICS | AM. COMPUTERS | AM. PHOTOS | COAUTHOR CS | COAUTHOR PHYSICS |
|---|---|---|---|---|---|
| SUPERVISED GCN | $77.19 \pm 0.12$ | $86.51 \pm 0.54$ | $92.42 \pm 0.22$ | $93.03 \pm 0.31$ | $95.65 \pm 0.16$ |
| GRACE | $\underline{80.14 \pm 0.48}$ | $89.53 \pm 0.35$ | $92.78 \pm 0.45$ | $91.12 \pm 0.20$ | – |
| BGRL | $79.98 \pm 0.10$ | $\underline{90.34 \pm 0.19}$ | $93.17 \pm 0.30$ | $\underline{93.31 \pm 0.13}$ | $\underline{95.69 \pm 0.05}$ |
| GRAPHMAE | $70.60 \pm 0.90$ | $86.28 \pm 0.07$ | $90.05 \pm 0.08$ | $87.70 \pm 0.04$ | $94.90 \pm 0.09$ |
| SP$^2$GCL | $79.42 \pm 0.19$ | $90.09 \pm 0.32$ | $\underline{93.23 \pm 0.26}$ | $92.61 \pm 0.24$ | $93.77 \pm 0.25$ |
| SSGE | $79.19 \pm 0.57$ | $89.05 \pm 0.58$ | $92.61 \pm 0.22$ | $93.06 \pm 0.41$ | $94.10 \pm 0.21$ |
| LAPLACEGNN (**ours**) | $\mathbf{82.34 \pm 0.38}$ | $\mathbf{92.30 \pm 0.23}$ | $\mathbf{95.70 \pm 0.15}$ | $\mathbf{94.86 \pm 0.16}$ | $\mathbf{96.72 \pm 0.11}$ |

maximization by optimizing perturbations $\boldsymbol{\delta}_t \in \mathcal{B}_\epsilon$, with $\mathcal{B}_\epsilon = \{\boldsymbol{\delta}_t : \|\boldsymbol{\delta}_t\|_\infty \leq \epsilon\}$ representing the $\ell_\infty$-bounded perturbation space. We employ a cosine similarity-based loss (Thakoor et al., 2021) that operates on the normalized representations, as shown in Appendix A.3.1. Gradient accumulation over $T$ steps ensures stable updates. This approach combines adversarial robustness, bootstrapped efficiency, and momentum stability while maintaining linear complexity.

## 4 Experiment

We conduct a comprehensive empirical evaluation of LaplaceGNN, demonstrating its effectiveness across diverse settings, including node classification, graph regression and classification, transfer learning, and adversarial attack settings. Our study spans a wide range of dataset scales and incorporates various recent graph encoder architectures, such as attentional, convolutional, autoencoder, and message-passing neural networks. Datasets are described and summarized in Appendix B.

### 4.1 Evaluation Protocol

For most tasks, we adopt the standard linear evaluation protocol for graphs (Velickovic et al., 2019b; You et al., 2020). This protocol involves training each graph encoder in a fully unsupervised manner to compute embeddings for each node. A simple linear model is then trained on these frozen embeddings using a logistic regression loss with $\ell_2$ regularization or a Ridge regressor, depending on the downstream task, without propagating gradients back to the graph encoder. In line with prior works (Suresh et al., 2021b; Thakoor et al., 2021), we use GCN for node prediction tasks and GIN for graph prediction tasks as the teacher encoder $g_\xi$ across all methods to demonstrate performance gains. All experiments are repeated 10 times, and we report the mean and standard deviation of the evaluation metrics. For OGB datasets (Hu et al., 2020), we evaluate the performance with their original feature extraction and following the original training/validation/test dataset splits. For TU datasets (Morris et al., 2020), we follow the standard protocols (Sun et al., 2019; Zhu et al., 2019) and report the mean 10-fold cross-validation accuracy along with the standard deviation over 5 runs. Detailed experimental configurations and baseline descriptions are provided in Appendix C.

### 4.2 Unsupervised Learning

Our experimental evaluation begins with unsupervised learning scenarios across multiple benchmark datasets, as shown in Tables 1 and 2. Our LaplaecGNN method demonstrates consistent improvements over existing approaches across various graph types and tasks.

For **node classification tasks**, LaplaceGNN shows remarkable improvements on several benchmark datasets, as depicted in Table 1. On WikiCS, our method establishes new state-of-the-art results with 82.34% accuracy, surpassing GRACE (80.14%) and BGRL (79.98%), and supervised GCN as well. As shown in Appendix C Table 9, on the challenging ogbn-arXiv dataset, our SSL method achieves 74.87% test accuracy, outperforming both BGRL (71.64%) and supervised GCN (71.74%). This demonstrates the effectiveness of our approach on larger-scale academic citation networks. Similarly, for the protein-protein interaction (PPI) dataset (Table 10), we achieve a Micro-F1 score of 75.73%, outperforming other self-supervised meth-

Table 2: Graph classification results on TU datasets. Results are reported by accuracy% and standard deviation over 10 random seed runs. The best and second-best results are shown in bold and underlined, respectively. For sake of space, this table has been shortened and the full version is available in Appendix C, Table 11.

| DATASET | MUTAG | PROTEINS | IMDB-B | IMDB-M | COLLAB | NCI1 |
|---|---|---|---|---|---|---|
| SUPERVISED GCN | 85.63 ± 5.84 | 64.25 ± 4.32 | 74.02 ± 3.35 | 51.94 ± 3.81 | 79.01 ± 1.81 | 80.21 ± 2.02 |
| INFOGRAPH | 89.01 ± 1.13 | 74.44 ± 0.31 | 73.02 ± 0.91 | 49.71 ± 0.51 | 70.62 ± 1.12 | 73.82 ± 0.71 |
| MVGRL | 89.72 ± 1.13 | - | 74.22 ± 0.72 | 51.21 ± 0.67 | 71.31 ± 1.21 | 75.02 ± 0.72 |
| INFOGCL | 90.62 ± 1.32 | - | 75.12 ± 0.92 | 51.41 ± 0.89 | 80.01 ± 1.32 | 79.81 ± 0.46 |
| GRAPHMAE | 88.12 ± 1.32 | 75.30 ± 0.52 | 75.52 ± 0.52 | 51.61 ± 0.66 | 80.33 ± 0.63 | 80.42 ± 0.35 |
| AUGMAE | 88.21 ± 1.02 | 75.83 ± 0.24 | 75.50 ± 0.62 | 51.83 ± 0.95 | 80.40 ± 0.52 | 80.11 ± 0.43 |
| LAPLACEGNN (**ours**) | **92.85 ± 0.74** | **80.52 ± 0.47** | **77.12 ± 0.32** | **52.44 ± 1.19** | **82.41 ± 0.42** | **82.21 ± 0.42** |

Table 3: Graph classification performance in transfer learning setting on molecular classification task. The metric is accuracy% and Micro-F1 for PPI. The best and second-best results are shown in bold and underlined, respectively. For sake of space, this table has been shortened and the full version is available in Appendix C, Table 13.

| DATASET | PRE-TRAIN | ZINC-2M | | | | PPI-360K |
|---|---|---|---|---|---|---|
| | FINE-TUNE | BBBP | Tox21 | HIV | TOXCAST | PPI |
| NO-PRE-TRAIN-GCN | | 65.83±4.52 | 74.03±0.83 | 75.34±1.97 | 63.43±0.61 | 64.80±1.03 |
| GRAPHCL | | 69.78±0.75 | 73.90±0.70 | 78.53±1.16 | 62.44±0.66 | 67.98±1.00 |
| GRAPHMAE | | 72.06±0.60 | 75.50±0.56 | 77.20±0.92 | 64.10±0.30 | 72.08±0.86 |
| AUGMAE | | 70.08±0.75 | 78.03±0.56 | 77.85±0.62 | 64.22±0.47 | 70.10±0.92 |
| SP$^2$GCL | | 68.72±1.53 | 73.06±0.75 | 78.15±0.43 | 65.11±0.53 | 72.11±0.74 |
| LAPLACEGNN (**ours**) | | **75.70±0.92** | **80.76±0.87** | **81.43±0.74** | **67.89±0.66** | **76.71±1.33** |

ods while maintaining a reasonable gap from supervised approaches, considering the 40% missing feature information in the dataset.

On molecular **graph classification** benchmarks (Table 2), our method achieves state-of-the-art performance in all datasets, outperforming state-of-the-art methods. Notably, we outperform both traditional kernel methods and recent GNN-based approaches, including supervised methods like GIN (Xu et al., 2018) and GAT Veličković et al. (2017). The performance gain is particularly significant compared to other unsupervised methods like InfoGCL (Xu et al., 2021) and InfoGraph (Sun et al., 2019), with improvements of 2.23% and 3.84% respectively on MUTAG. On PROTEINS, our method outperforms AugMAE (Wang et al., 2024) and GraphMAE (Hou et al., 2022) by 4.69% and 5.22%.

Further molecular property prediction experiments demonstrate the effectiveness of our augmentation strategies, as shown in Appendix C, Table 12. When applying perturbations at the encoder's first and last hidden layer, we achieve superior performance across all datasets: HIV, Tox21, ToxCast and BBBP. These results outperform both contrastive, spectral, and masked approaches like AD-GCL (Suresh et al., 2021a), SP$^2$GCL (Bo et al., 2024) and AugMAE (Wang et al., 2024).

## 4.3 Transfer Learning

The transfer learning capabilities of our method are evaluated through pre-training on large-scale molecular (ZINC-2M) and protein interaction (PPI-360K) datasets, followed by fine-tuning on smaller downstream tasks, as shown in Table 3. When pre-trained on ZINC-2M and fine-tuned on molecular property prediction tasks, our methods achieve significant improvements over the best previous methods: on BBBP ↑ 3.64% over GraphMAE, on Tox21 ↑ 2.73% over AugMAE, on HIV ↑ 2.90% over GraphCL (You et al., 2020), and on ToxCast ↑ 2.78% over SP$^2$GCL. These results suggest that our method learns more transferable molecular representations compared to existing approaches. Similarly, on protein-protein interaction networks, when

pre-trained on PPI-360K and fine-tuned on the PPI downstream task, our model achieves a consistent improvement of 3.60% Micro-F1 score, outperforming the previous best result from SP$^2$GCL.

### 4.4 Adversarial Stability

To evaluate the robustness of our method, we conduct experiments under adversarial attack scenarios on the Cora dataset, as detailed in Appendix C, Table 14. We test against four types of attacks (random, DICE (Waniek et al., 2018), GF-Attack (Chang et al., 2020), and Mettack (Gosch et al., 2024b) with different perturbation levels ($\sigma = 0.05$ and $\sigma = 0.2$, with $\sigma \times E$ flipped edges for a graph with $E$ edges). Our method, under clean conditions, achieves 88.44%accuracy, outperforming other self-supervised methods as well. More importantly, under various attack scenarios, our method maintains better performance: for random attacks with $\sigma = 0.2$ (which randomly removes edges), we achieve 88.23% accuracy, showing minimal degradation from clean performance; under DICE attacks with $\sigma = 0.2$, our method keeps 87.01% accuracy, outperforming the next best method by 2.71%; against GF-Attack with $\sigma = 0.2$, we achieve 87.00% accuracy, demonstrating strong resistance to gradient-based attacks. Most notably, under the strongest Mettack with $\sigma = 0.2$, our method achieves 74.07% accuracy, outperforming the previous best method by 4.15% and showing substantially better robustness compared to baseline GCN (31.04%). Extensive graph-attack experiments have been left as possible future work, being our focus mainly on enhancing self-supervised learning at this stage.

## 5 Complexity Analysis and Ablation Studies

We evaluate the impact of different centrality measures on performance, probing the flexibility of our framework to adapt to various graph structures and downstream tasks. Specifically, Table 4 shows a comparison of our proposed model with heuristic augmentation schemes on node classification benchmark. Augmentations like degree centrality, Pagerank centrality, and Katz centrality are denoted as DC, PC, and KC, respectively. Our LaplaceGNN-Std, the standard version of LaplaceGNN with learnable centrality weights parameters, achieves the highest gains in Coauthor-CS, Amazon-Computers and Amazon-Photos datasets, outperforming handcrafted augmentation schemes, such as those used in baseline methods like GCA and BGRL.

Moreover, Table 5 summarizes the performance of LaplaceGNN variants and baseline methods across molecular property prediction tasks. Again, LaplaceGNN-Std with learnable weights parameters achieves the best results across most tasks, particularly excelling on ToxCast, HIV, and BBBP datasets. Notably, LaplaceGNN-PC performs competitively, especially on Tox21 and ToxCast, indicating that PageRank centrality aligns well with these tasks.

Our approach represents a significant step toward eliminating hand-crafted augmentations while maintaining linear input scaling and adversarial robustness through the exploitation of Laplacian graph properties. The observed performance improvements validate our methodology, suggesting that future work should focus on efficiency optimizations by incorporating practical large-graph training techniques, as detailed in Section 6.

We break down here the complexity analysis into three components, that is, (1) the Laplacian centrality augmentations scheme (whose time and memory complexities are reported in Table 6), (2) the adversarial bootstrapped training, and (3) the memory consumption comparison.

**Laplacian Augmentations** This paragraph provides additional details on the Laplacian augmentation module, including its computational complexity, optimization strategies, and ablation studies analyzing the impact of centrality measures on downstream performance. We further validate the design choices of our spectral bootstrapping framework through empirical evaluations, as shown in Table 4 and Table 5.

The eigen-decomposition step for Laplacian optimization initially incurs $\mathcal{O}(Tn^3)$ time complexity. To mitigate this, we propose a selective eigen-decomposition method focusing on $K$ extremal eigenvalues, reducing complexity to $\mathcal{O}(TKn^2)$. Scalability improvements via sampling strategies (e.g., ego-nets) are deferred to future work.

Table 4: Centrality ablation studies with heuristic augmentation schemes on four node classification benchmark. Augmentations like degree centrality, PageRank centrality, and Katz centrality are denoted as DC, PC, and KC, respectively. For GCA, we report the numbers provided in their original paper.

| Method | Coauthor CS | Coauthor Physics | Am. Computers | Am. Photos |
|---|---|---|---|---|
| Supervised GCN | 93.03 ± 0.31 | 95.65 ± 0.16 | 86.51 ± 0.54 | 92.42 ± 0.22 |
| BGRL Standard | 93.31 ± 0.13 | 95.69 ± 0.05 | 90.34 ± 0.19 | 93.17 ± 0.30 |
| BGRL-DC | 93.34 ± 0.13 | 95.62 ± 0.09 | 90.39 ± 0.22 | 93.15 ± 0.37 |
| BGRL-PC | 93.34 ± 0.11 | 95.59 ± 0.09 | 90.45 ± 0.25 | 93.13 ± 0.34 |
| BGRL-EC | 93.32 ± 0.15 | 95.62 ± 0.06 | 90.20 ± 0.27 | 93.03 ± 0.39 |
| GCA Standard | 92.93 ± 0.01 | 95.26 ± 0.02 | 86.25 ± 0.25 | 92.15 ± 0.24 |
| GCA-DC | 93.10 ± 0.01 | 95.68 ± 0.05 | 87.85 ± 0.31 | 92.49 ± 0.09 |
| GCA-PC | 93.06 ± 0.03 | 95.69 ± 0.03 | 87.80 ± 0.23 | 92.53 ± 0.16 |
| GCA-EC | 92.95 ± 0.13 | 95.73 ± 0.03 | 87.54 ± 0.49 | 92.24 ± 0.21 |
| LaplaceGNN-DC (**ours**) | 93.11 ± 0.05 | 95.65 ± 0.11 | 90.04 ± 0.28 | 92.40 ± 0.15 |
| LaplaceGNN-PC (**ours**) | 93.40 ± 0.06 | 95.70 ± 0.03 | 91.12 ± 0.25 | 94.40 ± 0.10 |
| LaplaceGNN-KC (**ours**) | 93.52 ± 0.07 | 95.65 ± 0.10 | 90.70 ± 0.23 | 93.80 ± 0.23 |
| LaplaceGNN-Std (**ours**) | **94.86 ± 0.16** | **96.72 ± 0.11** | **92.30 ± 0.23** | **95.70 ± 0.15** |

Table 5: Centrality ablation studies on graph classification benchmark. Augmentations like degree centrality, PageRank centrality, and Katz centrality are denoted as DC, PC, and KC, respectively. Results are reported by accuracy% and standard deviation over 10 random seed runs.

| Dataset | HIV | Tox21 | ToxCast | BBBP |
|---|---|---|---|---|
| InfoGraph | 76.81 ± 1.01 | 69.74 ± 0.57 | 60.63 ± 0.51 | 66.33 ± 2.79 |
| GraphCL | 75.95 ± 1.35 | 72.40 ± 1.01 | 60.83 ± 0.46 | 68.22 ± 1.89 |
| MVGRL | 75.85 ± 1.81 | 70.48 ± 0.83 | 62.17 ± 1.61 | 67.24 ± 1.39 |
| JOAO | 75.37 ± 2.05 | 71.83 ± 0.92 | 62.48 ± 1.33 | 67.62 ± 1.29 |
| GMT | 77.56 ± 1.25 | 77.30 ± 0.59 | 65.44 ± 0.58 | 68.31 ± 1.62 |
| AD-GCL | 78.66 ± 1.46 | 71.42 ± 0.73 | 63.88 ± 0.47 | 68.24 ± 1.47 |
| GraphMAE | 77.92 ± 1.47 | 72.83 ± 0.62 | 63.88 ± 0.65 | 69.59 ± 1.34 |
| AugMAE | 78.37 ± 2.05 | 75.11 ± 0.69 | 62.48 ± 1.33 | 71.07 ± 1.13 |
| SP$^2$GCL | 77.56 ± 1.25 | 74.06 ± 0.75 | 65.23 ± 0.92 | 70.72 ± 1.53 |
| LaplaceGNN-DC (**ours**) | 78.43 ± 0.95 | 74.98 ± 0.83 | 66.04 ± 0.49 | 71.92 ± 1.03 |
| LaplaceGNN-PC (**ours**) | 79.70 ± 0.92 | 75.80 ± 0.79 | 66.31 ± 0.41 | 72.66 ± 0.98 |
| LaplaceGNN-KC (**ours**) | 78.68 ± 0.77 | 75.25 ± 0.67 | 65.98 ± 0.56 | 71.80 ± 1.14 |
| LaplaceGNN-Std(**ours**) | **80.83 ± 0.87** | **77.46 ± 0.70** | **67.74 ± 0.43** | **73.88 ± 1.10** |

For a graph with $n$ nodes, the Laplacian optimization scheme requires $T$ iterations with a time complexity of $\mathcal{O}(Tn^3)$ due to the eigen-decomposition operation $\lambda_{mod1,2}$, which becomes computationally prohibitive for large-scale graphs. To address this computational challenge, we propose a selective eigen-decomposition approach that focuses on $K$ lowest and highest eigenvalues using either the low-rank SVD or the Lanczos Algorithm (Parlett & Scott, 1979). This selective approach is theoretically justified as both extremal eigenvalues are fundamental to graph analysis and GNNs architecture design. Through this optimization, we achieve a reduced time complexity of $\mathcal{O}(TKn^2)$. The spectral divergence objective is approximated as:

$$\mathcal{L}(\Delta) \approx \pm \frac{1}{K} \sum_{k=1}^{K} \left( \lambda_{\text{mod}}^{(k)} - \lambda_{\text{orig}}^{(k)} \right)^2, \tag{3}$$

where $\lambda_{\text{orig}}^{(k)}$ and $\lambda_{\text{mod}}^{(k)}$ are eigenvalues of the original/modified Laplacians. Adjacency and augmentation matrices ($\mathbf{A}, \Delta_1, \Delta_2$) are stored in sparse format. Gradient updates use sparse-dense multiplications to minimize memory overhead. These optimizations ensure scalability while maintaining the theoretical guarantees of our max-min centrality-guided framework. For graphs large with $n \sim 10^4$, we subsample node pairs during Bernoulli sampling (Algorithm 1) with a sampling ratio $\rho \in [0, 1]$. A decaying learning rate $\eta_t = \eta_0/\sqrt{t}$ stabilizes convergence during gradient updates.

Further scalability improvements can be achieved through established practical treatments for large-scale graphs (Qiu et al., 2020), such as ego-nets sampling and batch training strategies, which we identify as promising directions for future research.

Table 6: Time and memory complexity of key components in LaplaceGNN, where $n_z$ are the non-zero entries in $\tilde{\mathbf{A}}$, while $E$ the total number of edges.

| COMPONENT | TIME COMPLEXITY | MEMORY COMPLEXITY |
|---|---|---|
| FULL EIGEN-DECOMPOSITION | $\mathcal{O}(Tn^3)$ | $\mathcal{O}(n^2)$ |
| SELECTIVE EIGEN-DECOMPOSITION | $\mathcal{O}(TKn^2)$ | $\mathcal{O}(Kn)$ |
| SPARSE LAPLACIAN UPDATES | $\mathcal{O}(Tn_z)$ | $\mathcal{O}(n_z)$ |

Table 7: Comparison of peak memory consumption (in GB) on standard node classification benchmarks, same as Table 1 plus the challenging ogbn-arXiv dataset. The best and second-best results are shown in bold and underlined, respectively. Symbol "−" indicates running out of memory on a 24GB GeForce RTX 4090 GPU.

| DATASET | AM. PHOTOS | WIKICS | AM. COMPUTERS | COAUTHOR CS | COAUTHOR PHYSICS | OGBN-ARXIV |
|---|---|---|---|---|---|---|
| # NODES | 7,650 | 11,701 | 13,752 | 18,333 | 34,493 | 169,343 |
| # EDGES | 119,081 | 216,123 | 245,861 | 81,894 | 247,962 | 1,166,243 |
| GRACE | 6.81 | 10.82 | 9.51 | 14.78 | − | − |
| BGRL | **2.19** | _7.41_ | _3.71_ | _4.47_ | **7.51** | _13.65_ |
| LAPLACEGNN | _2.85_ | **5.83** | **3.19** | **3.98** | _7.90_ | **11.34** |

**Adversarial Bootstrapped Training**   The adversarial bootstrapped training doesn't introduce any further memory requirements compared to other negative-free methods, therefore we rely on the explanation given by Thakoor et al. (2021), and compare it to previous popular contrastive methods such as GRACE (Zhu et al., 2020) and GCA (Zhu et al., 2021b) to show that it maintains a linear memory scaling. Indeed, LaplaceGNN reduces the memory complexity from $\mathcal{O}(N^2)$ to $\mathcal{O}(N)$ through bootstrapping training. Empirical memory consumption comparisons are detailed within the next paragraph.

Consider a graph with $V$ nodes and $E$ edges, and encoders that compute embeddings in time and space $\mathcal{O}(V + E)$. LaplaceGNN performs four encoder computations per update step (twice for both teacher and student networks, one for each augmented view) plus node-level prediction steps. The backward pass, approximately equal in cost to the forward pass, requires two additional encoder computations. Thus, the total complexity per update step for LaplaceGNN is $6\mathcal{C}_{\text{encoder}}(M + N) + 4\mathcal{C}_{\text{prediction}}N + \mathcal{C}_{\text{boot}}N$, where $\mathcal{C}_{\text{encoder}}$, $\mathcal{C}_{\text{prediction}}$, and $\mathcal{C}_{\text{boot}}$ are architecture-dependent constants. In contrast, both GRACE and GCA perform two encoder computations (one per augmentation) plus node-level projections, with two backward passes. Its total complexity is $4\mathcal{C}_{\text{encoder}}(M + N) + 4\mathcal{C}_{\text{projection}}N + \mathcal{C}_{\text{contrast}}N^2$. The crucial difference lies in the final term: while LaplaceGNN scales linearly with $N$ through bootstrapping, GRACE incurs quadratic cost from all-pairs contrastive computation. Empirical comparisons on benchmark datasets confirm these theoretical advantages, with detailed results presented in Table 7.

**Memory Consumption Comparison**   We conduct an empirical analysis of memory consumption across different self-supervised graph learning methods to validate our theoretical complexity claims. The comparison includes our LaplaceGNN against popular contrastive (GRACE) and bootstrapped (BGRL) baselines, as shown in Table 7.   This validates our theoretical analysis in Section 5 by eliminating negative samples LaplaceGNN achieves practical scalability while maintaining spectral learning capabilities. The memory savings become particularly significant on large real-world graphs, where LaplaceGNN maintains consistent memory efficiency across all scales, requiring 11.34GB for ogbn-arXiv (169k nodes) vs. BGRL's 13.65GB. On the other hand, GRACE fails on larger graphs due to quadratic contrastive costs, while LaplaceGNN/BGRL remain viable through linear bootstrapping.

## 6   Conclusion and Future Work

In this work, we introduced LaplaceGNN, a novel self-supervised learning framework for graph neural networks that leverages adversarial perturbations and bootstrapped aggregation to learn robust graph repre-

sentations. Our method advances the state-of-the-art by several key aspects: (1) a Laplacian augmentation centrality-guided module that generates views to avoid tuning handcrafted transformations, (2) an adversarial teacher-student network that ensures robust representations through bootstrapped knowledge transfer, and (3) a stable linear bootstrapping method that eliminates the quadratic cost needed for negative sampling. It demonstrates that carefully designed adversarial perturbations can serve as an effective form of data augmentation for self-supervised learning on graphs. The experimental results comprehensively validate our approach across diverse scenarios. In unsupervised learning tasks, we achieved and surpassed state-of-the-art performances on multiple benchmarks, including molecular property prediction, node classification, and graph classification tasks. Then, our method demonstrates superior transfer learning capabilities when pre-trained on large-scale datasets, with significant improvements over existing approaches on both molecular and protein interaction networks. Lastly, the robust performance under various adversarial attacks highlights the effectiveness of our adversarial bootstrapping strategy in learning stable and reliable representations.

Looking forward, several promising directions emerge for future research. The application of hierarchical graph pooling techniques (Ying et al., 2018; Bianchi & Lachi, 2023) could enhance our method's ability to capture multi-scale structural information. Integration with graph diffusion models (Chamberlain et al., 2021; Liu et al., 2023) presents an opportunity to better model long-range dependencies in large graphs. Further scalability improvements can be achieved through established practical treatments for large-scale graphs (Liao et al., 2022; Borisyuk et al., 2024), which we identify as promising directions for future research.

Future work will focus on optimizing memory usage for large-scale graphs, exploring more efficient training strategies, and investigating the theoretical foundations of adversarial bootstrapping in the context of graph neural networks. The integration of these various research directions could lead to even more powerful and practical self-supervised learning frameworks for real-world graph data.

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

# A    Further Discussion on LaplaceGNN Algorithm

LaplaceGNN is a self-supervised framework for graph representation learning that leverages spectral properties to learn robust features via adversarial bootstrapping training. Given an input graph $\mathcal{G} = (V, E)$ with adjacency matrix $\mathbf{A} \in \{0,1\}^{n \times n}$ and node features $\mathbf{X} \in \mathcal{R}^{n \times d}$, the goal is to learn representations capturing both structural and attribute information without manual augmentations or negative samples. LaplaceGNN comprises three key components: (1) a Laplacian augmentation module generating centrality-guided views, avoiding handcrafted transformations; (2) an adversarial online (teacher) network ensuring robust representations while the target (student) network learns via knowledge distillation; and (3) a linear bootstrapping method eliminating the quadratic cost of negative sampling. Laplacian augmentations and adversarial bootstrapping are detailed in Algorithm 1 and Algorithm 2, respectively.

## A.1    Centrality-Based Augmentation Scheme

Unlike fixed heuristic augmentations, we propose a principled augmentation method based on centrality measures and spectral graph theory, adaptable to diverse graph structures and tasks. The key idea is to maximize spectral differences while preserving critical structural properties through centrality-guided augmentations.

We introduce a centrality-guided augmentation scheme combining multiple centrality measures to capture node importance. Let $C(i)$ denotes the centrality score of node $i$, derived from a set of centrality functions $\mathcal{C} = \{C_1, C_2, \ldots, C_K\}$. Each $C_k$ corresponds to a centrality measure such as degree, PageRank, Katz centrality, or any other task-specific centrality measure relevant to the problem domain. The generalized centrality score is given by:

$$C(i) = \sum_{k=1}^{K} \alpha_k \cdot C_k(i), \tag{4}$$

where $\alpha_k$ are learnable weights that adjust the contribution of each centrality measure. The combined centrality matrix $\mathbf{C}$ is constructed from these scores, normalized to $[0, 1]$, and used to guide the augmentation process. This formulation ensures adaptability to diverse graphs and tasks while focusing modifications on nodes critical for preserving structural integrity and creating challenging augmented views.

Key advantages include: flexibility to integrate domain-specific centrality measures, a principled balance of global and local augmentations, and preservation of critical graph properties while introducing meaningful variations.

### A.1.1    Importance of Centrality-Based Approach

Altering key nodes generates more challenging and diverse augmented views, helping the model learn to maintain stable predictions even when critical structural components change. Additionally, this process enforces representation invariance by capturing persistent features despite significant topological modifications. However, there is an inherent trade-off: meaningful augmentations require substantial changes, while preserving essential graph properties demands stability. To balance this, we employ **spectral optimization** to **regulate** the extent of modifications. A budget constraint $B$ limits the total augmentation magnitude, while normalizing centrality scores prevents excessive alterations to any single node, as detailed in Algorithm 1.

Our approach ensures "controlled disruption", introducing impactful changes that challenge the model while maintaining global graph structure through spectral constraints. By focusing on important nodes, we generate stronger learning signals, leading to more robust node embeddings, improved generalization under structural variations, and feature extractors that better capture invariant properties.

## A.2    Laplacian Optimization

We formulate view generation as a Laplacian optimization problem that refines centrality-based probabilities, as outlined in Algorithm 1. The process begins by computing a combined centrality matrix $\mathbf{C}$, which encodes

node importance using multiple centrality metrics. This initialization is critical, as it provides a principled basis for generating augmentation matrices $\Delta_1$ and $\Delta_2$. The outer product $\mathbf{CC}^T$ ensures higher augmentation probabilities for edges connecting structurally important nodes. The core optimization then operates on the normalized graph Laplacian.

For a given adjacency matrix $\mathbf{A}$, we compute its normalized version $\tilde{\mathbf{A}} = \mathbf{D}^{-1/2}\mathbf{A}\mathbf{D}^{-1/2}$, where $\mathbf{D}$ is the degree matrix. The Laplacian is then defined as $\mathbf{L} = \mathbf{D} - \mathbf{A}$, and its normalized version as

$$\tilde{\mathbf{L}} = \mathbf{D}^{-1/2}\mathbf{L}\mathbf{D}^{-1/2} = \mathbf{I} - \tilde{\mathbf{A}}.$$

Our optimization framework aims to generate two complementary matrices, $\Delta_1$ and $\Delta_2$, such that the spectral properties of the augmented graphs maximize and minimize divergence from the original graph.

The optimization objective is defined through the spectral distance:

$$\mathcal{L}(\Delta) = \pm \frac{\|\lambda(\tilde{\mathbf{L}}) - \lambda(\tilde{\mathbf{L}}_{\mathrm{mod}})\|}{\|\lambda(\tilde{\mathbf{L}})\|}, \tag{5}$$

where $\lambda(\tilde{\mathbf{L}})$ represents the eigenvalues of the original normalized Laplacian and $\lambda(\tilde{\mathbf{L}}_{\mathrm{mod}})$ represents the eigenvalues of the modified Laplacian. Specifically, $\mathcal{L}(\Delta)$ corresponds to the max and min updates for the augmentation matrices and is therefore embedded within the Laplace optimization described in Appendix A.2.1.

### A.2.1 Two-View Generation

The algorithm generates two augmented views through sequential optimization:

1. **Maximization View ($\mathbf{A}_1$):**

   - Updates $\Delta_1$ using gradient ascent:

   $$\Delta_1^{(t+1)} \leftarrow \mathrm{Proj}_{[0,1],B}(\Delta_1^{(t)} + \eta \nabla_{\Delta_1} \|\lambda(\tilde{\mathbf{L}}_{\mathrm{mod1}})\|_2).$$

   - Pushes the spectral properties away from the original Laplacian.
   - Modified Laplacian computed as $\tilde{\mathbf{L}}_{\mathrm{mod1}} = \tilde{\mathbf{L}} + \tilde{\mathbf{L}}(C\Delta_1 C^T)$.

2. **Minimization View ($\mathbf{A}_2$):**

   - Updates $\Delta_2$ using gradient descent:

   $$\Delta_2^{(t+1)} \leftarrow \mathrm{Proj}_{[0,1],B}(\Delta_2^{(t)} - \eta \nabla_{\Delta_2} \|\lambda(\tilde{\mathbf{L}}_{\mathrm{mod2}})\|_2).$$

   - Maintains spectral properties close to the original Laplacian.
   - Modified Laplacian computed as $\tilde{\mathbf{L}}_{\mathrm{mod2}} = \tilde{\mathbf{L}} + \tilde{\mathbf{L}}(C\Delta_2 C^T)$.

This optimization is subject to two key constraints. First, **budget constraint**: the total number of augmentations is limited by $B = r \cdot (\sum_{i,j} \mathbf{A}_{ij})/2$, where $r$ is the budget ratio. Second, **structural preservation**: the centrality matrix $\mathbf{C}$ guides augmentations through the term $\mathbf{C}\Delta\mathbf{C}^T$, ensuring that modifications respect the underlying graph structure.

Then, the final augmented views are generated through Bernoulli sampling and the sampled $P_1, P_2$ are then applied to create the augmented Laplacian matrices:

$$\tilde{\mathbf{L}}_{\mathrm{mod1}} = \tilde{\mathbf{L}} + \tilde{\mathbf{L}}(C\Delta_1 C^T), \quad \tilde{\mathbf{L}}_{\mathrm{mod2}} = \tilde{\mathbf{L}} + \tilde{\mathbf{L}}(C\Delta_2 C^T). \tag{6}$$

### A.3 Self-Supervised learning via Adversarial Bootstrapping

We propose an adversarial bootstrapping framework that combines teacher-student knowledge transfer with adversarial training to learn robust graph representations. Unlike traditional contrastive methods relying on negative sampling, our approach leverages a bootstrapped learning scheme where the teacher network generates robust targets for the student network to learn from, eliminating the quadratic complexity of negative sampling (Thakoor et al., 2021).

We extend this approach by integrating a momentum encoder with an adversarially trained teacher network, introducing spectral constraints to ensure meaningful positive pairs while maintaining $O(N)$ in input memory complexity. Adversarial training enhances resistance to structural attacks, stabilizes gradients (by removing negative sampling), and improves transfer learning generalization. Our framework preserves memory efficiency by eliminating negative sampling while generating challenging yet meaningful positive pairs through adversarial perturbations. These improvements enable LaplaceGNN to scale linearly with input size while maintaining robustness against adversarial attacks, addressing key limitations in graph representation learning. The training scheme is detailed in Algorithm 2.

#### A.3.1 Adversarial Bootstrapping

Our framework incorporates adversarial training within a self-supervised learning context to enhance robustness and generalization without relying on handcrafted views informed by domain expertise. Given an input graph $\mathcal{G} = (V, E)$ with adjacency matrix $\mathbf{A} \in \{0, 1\}^{n \times n}$ and node features $\mathbf{X} \in \mathcal{R}^{n \times d}$, our objective is to learn an embedded representation $\mathbf{H} \in \mathbb{R}^{n \times d}$ to be used for downstream tasks, such as graph and node classification. For the sake of simplicity and notation's coherence, we adopt the node classification formalism throughout this section. However, our approach has been straightforwardly generalized to graph classification setup, as shown in Section 4.

Let $g_\xi$ denote the teacher encoder with parameters $\xi$, and $f_\theta$ denote the student encoder with parameters $\theta$. At each time $t$ we introduce adversarial perturbations $\boldsymbol{\delta}_t^{\mathbf{H}}$ on the hidden representations (or on initial node features as shown in Section 4) as

$$\mathbf{H} + \boldsymbol{\delta}_t^{\mathbf{H}}, \quad \boldsymbol{\delta}_t^{\mathbf{H}} \in \mathcal{B}_\epsilon, \tag{7}$$

where $\mathbf{H} = g_\xi(\hat{\mathcal{G}}_1, \mathbf{X})$ and $\mathcal{B}_\epsilon = \{\boldsymbol{\delta}_t : \|\boldsymbol{\delta}_t\|_\infty \leq \epsilon\}$ represents the $\ell_\infty$-bounded perturbation space. To keep the notation simple, we will use $\boldsymbol{\delta}_t$ rather than $\boldsymbol{\delta}_t^{\mathbf{H}}$ throughout this section. These perturbations are optimized through projected gradient descent:

$$\boldsymbol{\delta}_t = \text{Proj}_\epsilon(\boldsymbol{\delta}_{t-1} + \alpha \nabla_\delta \mathcal{L}_{\text{boot}}(\hat{\mathcal{G}}_1, \mathbf{H} + \boldsymbol{\delta}_{t-1})). \tag{8}$$

For the sake of space, we refer to Appendix A.3.2 for details on how the projector operator has been defined. The teacher-student framework is implemented through a dual-encoder architecture where the teacher (target) network and student (online) network share the same architectural blueprint but with the gradients flowing only through the teacher, as shown in Figure 1. The student network's parameters are updated through EMA of the teacher network's parameters and followed by projector $p_\theta$. To prevent representation collapse, the teacher network incorporates an additional MLP projector $q_\xi$, followed by a final MLP head. Therefore, LaplaceGNN's training procedure alternates between two phases, outer minimization through the bootstrapped loss $\mathcal{L}_{\text{boot}}$ and inner maximization by optimizing perturbations $\boldsymbol{\delta}_t \in \mathcal{B}_\epsilon$.

We employ a cosine similarity-based loss (Thakoor et al., 2021) that operates on the normalized representations:

$$\mathcal{L}_{\text{boot}} = -\frac{2}{N} \sum_{i=0}^{N-1} \frac{\langle \hat{\mathbf{T}}^{(i)}, \hat{\mathbf{Z}}^{(i)} \rangle}{\|\hat{\mathbf{T}}^{(i)}\| \|\hat{\mathbf{Z}}^{(i)}\|}, \tag{9}$$

where $\hat{\mathbf{T}} = \text{MLP}_\xi(q_\xi(\mathbf{H}^{(1)}))$ represents the teacher's projection and $\hat{\mathbf{Z}} = p_\theta(f_\theta(\hat{\mathcal{G}}_2, \mathbf{X}))$ represents the student's projection. To improve training efficiency while maintaining stability, we implement gradient accumulation over $T$ steps before parameter updates:

$$g_t = g_{t-1} + \frac{1}{T} \nabla_\xi \mathcal{L}_{\text{boot}}(\hat{\mathcal{G}}_1, \mathbf{H} + \boldsymbol{\delta}_{t-1}). \tag{10}$$

In this way, the complete training process alternates between optimizing adversarial perturbations $\delta$ and updating teacher and student parameters $\xi, \theta$ through accumulated gradients and exponential moving average (EMA), respectively. This approach combines the benefits of adversarial training (robustness), bootstrapped learning (efficiency), and momentum updates (stability) while maintaining linear computational complexity with respect to graph size.

### A.3.2 Projector Operator for Adversarial Bootstrapping

The operator $\mathrm{Proj}_\epsilon$ represents a projection that ensures the updated perturbation $\boldsymbol{\delta}_t$ stays within a specified constraint, typically a norm-ball of radius $\epsilon$. The projection is generally defined as follows:

$$
\mathrm{Proj}_\epsilon(\boldsymbol{\zeta}) = \begin{cases} \boldsymbol{\zeta}, & \text{if } \|\boldsymbol{\zeta}\|_2 \leq \epsilon \\ \epsilon \cdot \frac{\boldsymbol{\zeta}}{\|\boldsymbol{\zeta}\|_2}, & \text{otherwise} \end{cases}
$$

- For the $\ell_\infty$-norm, it clips the perturbation so that each component of $\boldsymbol{\zeta}$ is within the range $[-\epsilon, \epsilon]$.

- For the $\ell_2$-norm, it rescales the perturbation to ensure that its $\ell_2$-norm does not exceed $\epsilon$, effectively preventing the unbounded growth of $\boldsymbol{\zeta}$.

In our model configurations, we stick to the $\ell_\infty$-norm, which is generally the most common choice for projected gradient descent (PGD)-based adversarial training due to its simpler implementation and stronger robustness to adversarial attacks (since it bounds each perturbation dimension independently). In our model configurations, we stick to the $\ell_\infty$-norm, which is generally the most common choice for projected gradient descent (PGD)-based adversarial training due to its simpler implementation and stronger robustness to adversarial attacks (since it bounds each perturbation dimension independently).

## B  Datasets

We evaluate LaplaceGNN on a diverse set of benchmark datasets spanning node classification, graph classification, transfer learning, and adversarial attack tasks. Below we provide detailed descriptions of all datasets used in our experiments, following the evaluation protocol depicted in Section 4.1.

**Node Classification Datasets and Statistics**  The datasets used for node classification span various domains:

- WikiCS: A citation network of Wikipedia articles with links as edges and computer science categories as classes, with $11,701$ nodes, $216,123$ edges, $300$ features, and $10$ classes.

- Amazon Computers: A co-purchase network of Amazon products with $13,752$ nodes, $245,861$ edges, $767$ features, and $10$ classes.

- Amazon Photos: A co-purchase network of Amazon products with $7,650$ nodes, $119,081$ edges, $745$ features, and $8$ classess.

- Coauthor CS: An academic co-authorship network from the Microsoft Academic Graph with $8,333$ nodes, $81,894$ edges, $6,805$ features, and $15$ classes.

- Coauthor Physics: An academic co-authorship network with $34,493$ nodes, $247,962$ edges, $8,415$ features, and $5$ classes.

- ogbn-arXiv: A large-scale citation network from arXiv with paper subject categories with $169,343$ nodes, $1,166,243$ edges, $128$ features, and $40$ classes.

- Cora: A classic citation network with paper topics as classes with $2,708$ nodes, $5,429$ edges, $1,433$ features, and $7$ classes.

**Graph Classification and Statistics** The datasets used for graph classification are:

- MUTAG A chemical dataset for mutagenicity classification with 188 graphs, avg. 17.9 nodes, avg. 19.8 edges, and 2 classes.

- PROTEINS: A bioinformatics dataset for enzyme classification with $1,113$ graphs, avg. 39.1 nodes, avg. 72.8 edges, and 2 classes.

- IMDB-B: A social dataset representing movie collaborations with $1,000$ graphs, avg. 19.8 nodes, avg. 96.5 edges, and 2 classes.

- IMDB-M A social dataset with multi-class movie collaboration classification with $1,500$ graphs, avg. 13.0 nodes, avg. 65.9 edges, and 3 classes.

- COLLAB: A social dataset classifying collaboration types $5,000$ graphs, avg. 74.5 nodes, avg. 2457.8 edges, and 3 classes.

- NCI1: A chemical dataset for anticancer activity prediction with $4,110$ graphs, avg. 29.8 nodes, avg. 32.3 edges, and 2 classes.

- Tox21: A molecular dataset for toxicity prediction with $7,831$ graphs, avg. 18.6 nodes, avg. 38.6 edges, and 12 classes.

- HIV: A molecular dataset for HIV inhibition prediction with $41,127$ graphs, avg. 25.5 nodes, avg. 54.9 edges, and 2 classes.

- ToxCast: A molecular dataset containing toxicity assays with $8,576$ graphs, avg. 18.8 nodes, avg. 38.5 edges, and 617 classes.

- BBBP: A molecular dataset for permeability prediction with $2,039$ graphs, avg. 24.1 nodes, avg. 51.9 edges and 2 classes.

**Transfer Learning Datasets and Statistics** The datasets involved for transfer learning are:

- ZINC-2M: A large-scale molecular dataset with synthetic molecules used for pre-training in quantum property prediction with $2,000,000$ graphs.

- PPI-360K: A biological dataset of protein interaction networks across species for biological representation learning with $360,000$ graphs.

## C Experiment Configuration and Baseline Comparison

This appendix provides the complete list of tables (from Table 8 to Table 14) that have been shortened throughout Section 4 because of space constraints. The "−" symbol indicates running out of memory on a 24GB GeForce RTX 4090 GPU. Our experiments employ a two-layer GCN with 512 and 256 hidden units as the teacher encoder for node classification tasks and a three-layer GIN (512 hidden units per layer) for graph classification, following the architecture conventions of BGRL. The hidden size of the predictor is 512. The student network mirrors the teacher's architecture, with parameters updated via exponential moving averaging (EMA) at a momentum rate of 0.998. Training uses the Adam optimizer with a learning rate of $1e-5$, a weight decay of $1e-5$, and epochs of 5,000. For adversarial perturbations, we set the step size $\epsilon = 0.008$, $\delta = 8e-3$, accumulation steps $= 2$, lr $= 1e-5$, weight decay $= 8e-4$, and $T = 3$ PGD steps during inner maximization. Centrality-guided augmentations use a budget ratio $r = 0.5$ and selective eigendecomposition for the top/bottom $K = 100$ eigenvalues. Additional hyperparameters align with established protocols from BGRL (Thakoor et al., 2021), GraphMAE (Hou et al., 2022), and SP$^2$GCL (Bo et al., 2024) unless stated otherwise.

Table 8: Node classification performance measured by accuracy%, with standard deviations over 10 random seed runs. The best and second-best results are shown in bold and underlined, respectively. The "−" symbol indicates running out of memory on a 24GB GeForce RTX 4090 GPU.

| DATASET | WIKICS | AM. COMPUTERS | AM. PHOTOS | COAUTHOR CS | COAUTHOR PHYSICS |
|---|---|---|---|---|---|
| SUPERVISED GCN (2016) | 77.19 ± 0.12 | 86.51 ± 0.54 | 92.42 ± 0.22 | 93.03 ± 0.31 | 95.65 ± 0.16 |
| DGI (2019A) | 75.35 ± 0.14 | 83.95 ± 0.47 | 91.61 ± 0.22 | 92.15 ± 0.63 | 94.51 ± 0.52 |
| GMI (2020) | 74.85 ± 0.08 | 82.21 ± 0.31 | 90.68 ± 0.17 | – | – |
| MVGRL (2020) | 77.52 ± 0.08 | 87.52 ± 0.11 | 91.74 ± 0.07 | 92.11 ± 0.12 | 95.33 ± 0.03 |
| GRACE (2020) | 80.14 ± 0.48 | 89.53 ± 0.35 | 92.78 ± 0.45 | 91.12 ± 0.20 | – |
| BGRL (2021) | 79.98 ± 0.10 | 90.34 ± 0.19 | 93.17 ± 0.30 | 93.31 ± 0.13 | 95.69 ± 0.05 |
| GCA (2021B) | 78.35 ± 0.05 | 88.94 ± 0.15 | 92.53 ± 0.16 | 93.10 ± 0.01 | 95.03 ± 0.03 |
| GRAPHMAE (2022) | 70.60 ± 0.90 | 86.28 ± 0.07 | 90.05 ± 0.08 | 87.70 ± 0.04 | 94.90 ± 0.09 |
| AUGMAE (2024) | 71.70 ± 0.60 | 85.68 ± 0.06 | 89.44 ± 0.11 | 84.61 ± 0.22 | 91.77 ± 0.15 |
| SP$^2$GCL (2024) | 79.42 ± 0.19 | 90.09 ± 0.32 | 93.23 ± 0.26 | 92.61 ± 0.24 | 93.77 ± 0.25 |
| SSGE (2025B) | 79.19 ± 0.57 | 89.05 ± 0.58 | 92.61 ± 0.22 | 93.06 ± 0.41 | 94.10 ± 0.21 |
| LAPLACEGNN (**ours**) | **82.34 ± 0.38** | **92.30 ± 0.23** | **95.70 ± 0.15** | **94.86 ± 0.16** | **96.72 ± 0.11** |

Table 9: Performance on the ogbn-arXiv task measured in terms of classification accuracy along with standard deviations. Our experiments are averaged over 10 random model initializations. The best and second-best results are shown in bold and underlined, respectively. The "−" symbol indicates running out of memory on a 24GB GeForce RTX 4090 GPU.

| METHOD | VALIDATION ACCURACY | TEST ACCURACY |
|---|---|---|
| MLP | 57.65 ± 0.12 | 55.50 ± 0.23 |
| NODE2VEC (2016) | 71.29 ± 0.13 | 70.07 ± 0.13 |
| DGI (2019A) | 71.26 ± 0.11 | 70.34 ± 0.16 |
| MVGRL (2020) | – | – |
| GRACE-SUBSAMPLING (κ = 2048) (2020) | 72.61 ± 0.15 | 71.51 ± 0.11 |
| BGRL (2021) | 72.53 ± 0.09 | 71.64 ± 0.12 |
| INFOGCL (2021) | – | – |
| GRAPHMAE (2022) | 72.80 ± 0.23 | 71.30 ± 0.40 |
| AUGMAE (2024) | 73.11 ± 0.144 | 71.59 ± 0.20 |
| SSGE (2025B) | 73.09 ± 0.15 | 71.37 ± 0.19 |
| LAPLACEGNN (**ours**) | **76.39 ± 0.07** | **74.87 ± 0.23** |
| SUPERVISED GCN | 73.00 ± 0.17 | 71.74 ± 0.29 |

Table 10: Performance on the PPI inductive task on multiple graphs, measured in terms of Micro-F1 across the 121 labels along with standard deviations. Our experiments are averaged over 10 random model initializations. The best and second-best results are shown in bold and underlined, respectively. The gap between the best self-supervised methods and fully supervised methods is due to 40% of the nodes missing feature information. The "−" symbol indicates running out of memory on a 24GB GeForce RTX 4090 GPU.

| METHOD | MICRO-F1 (TEST) |
|---|---|
| RANDOM-INIT | 62.60 ± 0.20 |
| RAW FEATURES | 42.20 |
| DGI (2019A) | 63.80 ± 0.20 |
| GMI (2020) | 65.00 ± 0.02 |
| MVGRL (2020) | – |
| GRACE MEANPOOLING ENCODER (2020) | 69.66 ± 0.15 |
| BGRL MEANPOOLING ENCODER (2021) | 69.41 ± 0.15 |
| GRACE GAT ENCODER | 69.71 ± 0.17 |
| BGRL GAT ENCODER | 73.61 ± 0.15 |
| INFOGCL (2021) | – |
| GRAPHMAE (2022) | 74.10 ± 0.40 |
| AUGMAE (2024) | 74.30 ± 0.21 |
| LAPLACEGNN (**ours**) | **75.73 ± 0.16** |
| SUPERVISED GCN MEANPOOLING | **96.90 ± 0.20** |

Table 11: Graph classification results on TU datasets. Results are reported by accuracy% and standard deviation over 10 random seed runs. The best and second-best results are shown in bold and underlined, respectively. The "−" symbol indicates running out of memory on a 24GB GeForce RTX 4090 GPU.

| Dataset | MUTAG | PROTEINS | IMDB-B | IMDB-M | COLLAB | NCI1 |
|---|---|---|---|---|---|---|
| **Supervised GNN Methods** | | | | | | |
| GraphSAGE (2017) | 85.12 ± 7.62 | 63.95 ± 7.73 | 72.30 ± 5.32 | 50.91 ± 2.20 | − | 77.72 ± 1.50 |
| GCN (2016) | 85.63 ± 5.84 | 64.25 ± 4.32 | 74.02 ± 3.35 | 51.94 ± 3.81 | 79.01 ± 1.81 | 80.21 ± 2.02 |
| GIN-0 (2018) | 89.42 ± 5.61 | 64.63 ± 7.04 | 75.12 ± 5.14 | 52.32 ± 2.82 | 80.20 ± 1.92 | 82.72 ± 1.71 |
| GIN-$\varepsilon$ | 89.01 ± 6.01 | 63.71 ± 8.23 | 74.35 ± 5.13 | 52.15 ± 3.62 | 80.11 ± 1.92 | 82.71 ± 1.64 |
| GAT (2017) | 89.45 ± 6.14 | 66.73 ± 5.14 | 70.51 ± 2.31 | 47.81 ± 3.14 | − | − |
| **Unsupervised Methods** | | | | | | |
| Random Walk | 83.71 ± 1.51 | 57.91 ± 1.32 | 50.71 ± 0.31 | 34.72 ± 0.29 | − | − |
| node2vec (2016) | 72.61 ± 10.21 | 58.61 ± 8.03 | − | − | − | 54.92 ± 1.62 |
| sub2vec (2018) | 61.11 ± 15.81 | 60.03 ± 6.41 | 55.32 ± 1.52 | 36.71 ± 0.82 | − | 52.82 ± 1.53 |
| graph2vec (2017) | 83.22 ± 9.62 | 73.30 ± 2.05 | 71.12 ± 0.53 | 50.42 ± 0.91 | − | 73.22 ± 1.83 |
| InfoGraph (2019) | 89.01 ± 1.13 | 74.44 ± 0.31 | 73.02 ± 0.91 | 49.71 ± 0.51 | 70.62 ± 1.12 | 73.82 ± 0.71 |
| MVGRL (2020) | 89.72 ± 1.13 | - | 74.22 ± 0.72 | 51.21 ± 0.67 | 71.31 ± 1.21 | 75.02 ± 0.72 |
| GraphCL (2020) | 86.82 ± 1.32 | − | 71.11 ± 0.41 | − | 71.38 ± 1.12 | 77.81 ± 0.74 |
| GCC (2020) | 86.41 ± 0.52 | 58.41 ± 1.22 | 71.92 ± 0.52 | 48.91 ± 0.81 | 75.22 ± 0.32 | 66.92 ± 0.21 |
| JOAO (2021) | 87.31 ± 1.21 | 74.55 ± 0.43 | 70.22 ± 3.01 | 49.21 ± 0.70 | 69.51 ± 0.31 | 78.12 ± 0.47 |
| InfoGCL (2021) | 90.62 ± 1.32 | - | 75.12 ± 0.92 | 51.41 ± 0.89 | 80.01 ± 1.32 | 79.81 ± 0.46 |
| GraphMAE (2022) | 88.12 ± 1.32 | 75.30 ± 0.52 | 75.52 ± 0.52 | 51.61 ± 0.66 | 80.33 ± 0.63 | 80.42 ± 0.35 |
| AugMAE (2024) | 88.21 ± 1.02 | 75.83 ± 0.24 | 75.50 ± 0.62 | 51.83 ± 0.95 | 80.40 ± 0.52 | 80.11 ± 0.43 |
| SSGE (2025b) | 86.21 ± 0.92 | 71.25 ± 0.85 | 73.42 ± 0.32 | 48.71 ± 0.69 | 78.31 ± 0.72 | 77.81 ± 0.52 |
| LaplaceGNN (**ours**) | **92.85 ± 0.74** | **80.52 ± 0.47** | **77.12 ± 0.32** | **52.44 ± 1.19** | **82.41 ± 0.42** | **82.21 ± 0.42** |

Table 12: Graph classification results on OGB molecular datasets. Results are reported by accuracy% and standard deviation over 10 random seed runs. The best and second-best results are shown in bold and underlined, respectively.

| Dataset | HIV | Tox21 | ToxCast | BBBP |
|---|---|---|---|---|
| InfoGraph (2019) | 76.81 ± 1.01 | 69.74 ± 0.57 | 60.63 ± 0.51 | 66.33 ± 2.79 |
| GraphCL (2020) | 75.95 ± 1.35 | 72.40 ± 1.01 | 60.83 ± 0.46 | 68.22 ± 1.89 |
| MVGRL (2020) | 75.85 ± 1.81 | 70.48 ± 0.83 | 62.17 ± 1.61 | 67.24 ± 1.39 |
| JOAO (2021) | 75.37 ± 2.05 | 71.83 ± 0.92 | 62.48 ± 1.33 | 67.62 ± 1.29 |
| GMT (2021) | 77.56 ± 1.25 | 77.30 ± 0.59 | 65.44 ± 0.58 | 68.31 ± 1.62 |
| AD-GCL (2021a) | 78.66 ± 1.46 | 71.42 ± 0.73 | 63.88 ± 0.47 | 68.24 ± 1.47 |
| GraphMAE (2022) | 77.92 ± 1.47 | 72.83 ± 0.62 | 63.88 ± 0.65 | 69.59 ± 1.34 |
| AugMAE (2024) | 78.37 ± 2.05 | 75.11 ± 0.69 | 62.48 ± 1.33 | 71.07 ± 1.13 |
| SP$^2$GCL (2024) | 77.56 ± 1.25 | 74.06 ± 0.75 | 65.23 ± 0.92 | 70.72 ± 1.53 |
| LaplaceGNN (**ours**) | **80.83 ± 0.87** | **77.46 ± 0.70** | **67.74 ± 0.43** | **73.88 ± 1.10** |

Table 13: Graph classification performance in transfer learning setting on molecular classification task. The metric is accuracy% and Micro-F1 for PPI. The best and second-best results are shown in bold and underlined, respectively.

| Dataset | Pre-Train | ZINC-2M | | | | PPI-360K |
|---|---|---|---|---|---|---|
| | Fine-Tune | BBBP | Tox21 | HIV | ToxCast | PPI |
| No-Pre-Train-GCN | | 65.83±4.52 | 74.03±0.83 | 75.34±1.97 | 63.43±0.61 | 64.80±1.03 |
| InfoGraph | (2019) | 68.84±0.81 | 75.32±0.52 | 76.05±0.72 | 62.74±0.64 | 64.13±1.03 |
| GraphCL | (2020) | 69.78±0.75 | 73.90±0.70 | 78.53±1.16 | 62.44±0.66 | 67.98±1.00 |
| MVGRL | (2020) | 69.08±0.52 | 74.50±0.68 | 77.13±0.61 | 62.64±0.56 | 68.78±0.71 |
| AD-GCL | (2021a) | 70.08±1.15 | 76.50±0.80 | 78.33±1.06 | 63.14±0.76 | 68.88±1.30 |
| JOAO | (2021) | 71.48±0.95 | 74.30±0.68 | 77.53±1.26 | 63.24±0.56 | 64.08±1.60 |
| GraphMAE | (2022) | 72.06±0.60 | 75.50±0.56 | 77.20±0.92 | 64.10±0.30 | 72.08±0.86 |
| AugMAE | (2024) | 70.08±0.75 | 78.03±0.56 | 77.85±0.62 | 64.22±0.47 | 70.10±0.92 |
| SP$^2$GCL | (2024) | 68.72±1.53 | 73.06±0.75 | 78.15±0.43 | 65.11±0.53 | 72.11±0.74 |
| LaplaceGNN (**ours**) | | **75.70±0.92** | **80.76±0.87** | **81.43±0.74** | **67.89±0.66** | **75.71±1.33** |

Table 14: Node classification performance on Cora in adversarial attack setting measured by accuracy%. The best and second-best results are shown in bold and underlined, respectively.

| Attack Method | Clean | Random | | DICE | | GF-Attack | | Mettack | |
|---|---|---|---|---|---|---|---|---|---|
| | $\sigma = 0$ | $\sigma = 0.05$ | $\sigma = 0.2$ | $\sigma = 0.05$ | $\sigma = 0.2$ | $\sigma = 0.05$ | $\sigma = 0.2$ | $\sigma = 0.05$ | $\sigma = 0.2$ |
| Supervised GCN | 81.34±0.35 | 81.11±0.32 | 80.02±0.36 | 79.42±0.37 | 78.37±0.42 | 80.12±0.33 | 79.43±0.32 | 50.29±0.41 | 31.04±0.48 |
| GRACE (2020) | 83.33±0.43 | 83.23±0.38 | 82.57±0.48 | 81.28±0.39 | 80.72±0.44 | 82.59±0.35 | 80.23±0.38 | 67.42±0.59 | 55.26±0.53 |
| BGRL (2021) | 83.63±0.38 | 83.12±0.34 | 83.02±0.39 | 82.83±0.48 | 81.92±0.39 | 82.10±0.37 | 80.98±0.42 | 70.23±0.48 | 60.42±0.54 |
| GBT (2022) | 80.24±0.42 | 80.53±0.39 | 80.20±0.35 | 80.32±0.32 | 80.20±0.34 | 79.89±0.41 | 78.25±0.49 | 63.26±0.69 | 53.89±0.55 |
| MVGRL (2020) | 85.16±0.52 | 86.29±0.52 | 86.21±0.78 | 83.78±0.35 | 83.02±0.40 | 83.79±0.39 | 82.46±0.52 | 73.43±0.53 | 61.49±0.56 |
| GCA (2021b) | 83.67±0.44 | 83.33±0.46 | 82.49±0.37 | 82.20±0.32 | 81.82±0.45 | 81.83±0.36 | 79.89±0.47 | 58.25±0.68 | 49.25±0.62 |
| GMI (2020) | 83.02±0.33 | 83.14±0.38 | 82.12±0.44 | 82.42±0.44 | 81.13±0.49 | 82.13±0.39 | 80.26±0.48 | 60.59±0.54 | 53.67±0.68 |
| DGI (2019a) | 82.34±0.64 | 82.10±0.58 | 81.03±0.52 | 85.52±0.59 | 84.30±0.63 | 81.30±0.54 | 79.88±0.58 | 71.42±0.63 | 63.93±0.58 |
| SP²GCL (2024) | 85.86±0.57 | 85.28±0.49 | 84.21±0.42 | 80.48±0.38 | 79.89±0.43 | 85.08±0.77 | 84.28±0.82 | 77.28±0.82 | 69.92±0.83 |
| LaplaceGNN (ours) | **88.44±0.71** | **88.83±0.47** | **88.23±0.38** | **88.20±0.25** | **87.01±0.46** | **88.11±0.87** | **87.00±0.74** | **80.08±0.91** | **74.07±0.70** |

