# OpenReview forum: "Self-Supervised Graph Learning via Spectral Bootstrapping and Laplacian-Based Augmentations"
_TMLR — Rejected by TMLR_

### Review · Reviewer_jrks · 2025-10-30

**Summary Of Contributions:**

This paper introduces an unsupervised technique for graph representation learning, which the authors describe as distinct from contrastive learning approaches that rely on negative sampling. Instead, the proposed method trains a graph neural network (GNN) in a teacher–student framework to be robust to “adversarial” perturbations obtained by maximizing a notion of spectral divergence. The main claimed difference from contrastive methods is that the proposed approach does not require quadratic complexity—although this is not necessarily the case for all contrastive methods.

The authors argue that the method is:

- robust to adversarial perturbations
- transferable to different domains
- independent of handcrafted augmentations
- simpler
- more efficient
- more performant across a wide range of tasks

My overall assessment is as follows:

- the main paper is often unclear and not entirely self-contained
- the method does not appear simpler than a general contrastive learning approach
- computational complexity and wall-clock time are not compared to those of contrastive methods, which would be necessary to support the efficiency claims
- the description of the evaluation procedures is unclear and difficult to replicate; additionally, the unsupervised training process appears to include test data (inputs only), which may result in overly favorable outcomes
- no explicit hyperparameter tuning appears to have been performed for the baselines, which could lead to underestimating their performance; it is also unclear how hyperparameters were selected for the proposed approach. Citing “previously established choices” does not constitute sufficient justification.

**Additional Comments:**

Why do the authors need a minimization of the spectral divergence? Can the original graph be used instead? Could you please show us what happens if the minimization objective is not used?

**Audience:**

No

**Audience Explanation:**

I find the main paper somewhat unclear to read and in need of a substantial revision of the text. If we focus less on the results and more on the underlying idea, the motivation appears relatively weak and the methodology is intuitively difficult to follow. Mentioning adversarial attacks in the context of drug development seems rather unrealistic, given that pharmaceutical companies typically do not expose their models or data to external parties.

There does not appear to be a clear conceptual difference between the adversarial nature of their approach and the contrastive nature of others, yet the authors claim that contrastive methods are generally problematic as they require a quadratic number of negative samples—a claim that has not been supported by adequate evidence.

Some symbols are undefined: K in Table 6, EMA, symbols g, q, p in Algorithm 2, $\Delta_{1,2}$, etc.

Several concepts are introduced before being defined, starting from an out-of-context “negative sampling” term in the first sentence of the abstract. Still in the abstract, “linear scaling” is ambiguous, “more efficient” lacks the object of comparison, and “strengthens feature learning” uses unclear terminology. Generally speaking, the authors should revise their text to ensure that (i) terminology is introduced only after being properly defined, and (ii) the object of comparison is always specified. It is also not clear why a teacher–student architecture is needed.

I am not convinced that the TMLR audience would find this paper particularly compelling, because while the results look promising (without considering my serious concerns about the empirical evaluation), it remains largely unclear why the proposed approach should work as described.

**Broader Impact Concerns:**

No concerns.

**Claims And Evidence:**

No

**Claims Explanation:**

The evidence supporting the claims consists primarily of quantitative results on several datasets.

Regarding the computational complexity claims, there is no clear evidence that contrastive learning approaches inherently have quadratic complexity. If the authors are referring to specific contrastive learning methods, this should be stated explicitly. In general, the number of negative samples does not necessarily have to scale quadratically with the number of nodes.

The proposed method does not appear to be simpler than a general contrastive learning approach, which typically involves generating perturbed graphs. The proposed approach also generates perturbations via “Laplacian spectral divergence,” so it is unclear why it should be considered simpler.

All claims of transferability and performance rely on substantially higher results across all tasks presented. This raises concerns about whether the evaluation was fair and robust. In this regard, Section 4.1 contains insufficiently clear information on how the results can be replicated. Early stopping (i.e., selecting the best epoch based on validation accuracy) is not reported in the paper, although it is used in the implementation. It is also unclear when experiments were repeated 10 times and when 5, and whether these repetitions refer to model selection or risk assessment.

The 10-fold CV evaluation, as used in prior work such as GIN, has been previously critiqued, so further clarification is needed on the evaluation setup. As noted, the lack of precise information about the evaluation process significantly hinders replicability. Model selection (i.e., hyperparameter tuning) does not appear to have been performed for any of the baselines, which could result in underestimating their performance. It is also unclear how the hyperparameters were chosen, and why the test performance was used to update the progress bar during training, since test accuracy should typically only be examined after selecting the best validation epoch.

For these reasons, the presented results may not accurately reflect the true risk of each model. The fact that the proposed model performs best on all tasks raises potential concerns about cherry-picking or unfair evaluation practices, which requires clarification.

Finally, the claim that the method does not rely on handcrafted perturbation strategies does not seem accurate. As shown in Equation 1, the method depends on user-defined centrality metrics, and this is further evidenced by Tables 4 and 5.

**Requested Changes:**

- Please provide quantitative evidence demonstrating why contrastive learning approaches require quadratic complexity in the number of nodes to achieve the target performance, or cite related works supporting that this is the case for most methods. Alternatively, consider revising your narrative and statements to clarify if this holds only for a selected subset of models.

- Please clarify in Section 4.1 how hyperparameters were selected for all models and baselines. Appropriate hyperparameter tuning should be performed for all baselines, unless results obtained under the same experimental setup are already available.

- Please elaborate on the key conceptual differences between your adversarial approach and contrastive approaches. Since both rely on some form of perturbation, it would be helpful to clearly explain what distinguishes your method from contrastive ones.

---

### Review · Reviewer_Ndem · 2025-11-02

**Summary Of Contributions:**

The paper studies self-supervised graph representation learning and proposed the LaplacianGNN framework as a negative-sample-free method of learning with spectral augmentation. The authors claimed that the primary algorithmic innovations are two-fold:
- LaplacianGNN utilizes optimization techniques in the graph spectral domain to create contrasting views as a principled method of building augmentations.
- LaplacianGNN devices a adversarial bootstrapping that uses a teacher-student architecture to facilitate contrastive training.

Empirical validations are conducted over benchmark datasets showcasing the efficacy of the proposed framework.

**Additional Comments:**

[1]. Lin, Lu, Jinghui Chen, and Hongning Wang. "Spectral augmentation for self-supervised learning on graphs." arXiv preprint arXiv:2210.00643 (2022).
[2]. Caron, Mathilde, et al. "Emerging properties in self-supervised vision transformers." Proceedings of the IEEE/CVF international conference on computer vision. 2021.

**Audience:**

Yes

**Audience Explanation:**

Self-supervised GNN learning is an important problem in the graph representation learning community. The paper focuses on the critical issue of augmentation construction which could be of interest to TMLR's audience.

**Claims And Evidence:**

No

**Claims Explanation:**

The primary concern is novelty: Both innovation claims in this paper seems to be closely related to previous developments. Specifically:
- ``Spectral augmentation``: The spectral augmentation method in this paper seems to coincide pretty much with [1]. For example, both works adopt optimization techniques in the spectral domain which almost take the same form, as well as using minimization and maximization as construction routines. The only difference seems to be the additional incorporation of centrality measures proposed in LaplacianGNN, which personally I think is not a major novelty.
- ``Adversarial bootstrapping``: The adversarial bootstrapping method in this paper, which uses teacher-student architectures seem to be closely related to standard techniques in computer vision domain such as DINO [2], I don't quite get the essential innovations here.

Based on the above evidence, I do think that the paper lacks novelty.

**Requested Changes:**

Mainly two:
- The authors should compare their approach with SPAN in [1], highlighting essential algorithmic differences and demonstrate performance comparisons.
- The authors should state clearly the  essential novelty of the adversarial bootstrapping method, and discuss connections to self-supervised representation learning techniques.

---

### Review · Reviewer_3fRu · 2025-12-09

**Summary Of Contributions:**

This work focuses on the problem of self-supervised learning on graphs. While many methods for self-supervised learning on graphs have been developed in the last five years, few of them are easy to deploy in practice as (a) they rely on augmentation schemes that are "handcrafted" and that are difficult to generalize from dataset to dataset, and (b) a lot of these approaches are not robust to adversarial perturbation on graphs.

The contributions of the authors are two-fold:
(a) they suggest an augmentation scheme based on a modification of the laplacian. More precisely, they compute centrality scores, which they combine in a score with learnable weights on each of the centralities. Then, starting with $\Delta_1 =CC^T$, they progressively update it to maximize the score" $\tilde{L}  + \tilde{L} C \Delta_1 C^T$. A second laplacian is computed by minimizing that same score. The final augmented views are generated through Bernoulli sampling of $\Delta_1$ and $\Delta_2$.
(b) An adversarial bootstrapping techniques that doesnt rely on negative pairs for self supervision.

The validity of the method is supported by a set of experiments on a number of different tasks.

**Audience:**

Yes

**Audience Explanation:**

This is a reasonably interesting addition to the literature --- I am sympathetic to the point made by the authors that a lot of self-supervised methods rely on many hyperparameters that are difficult to tune, so their method could be promising, as it seems to yield interesting performance gains. However, in the current form, the paper is unclear on key components of the method, and lacking proper ablation studies that would  validate the benefits of their approach.

**Claims And Evidence:**

No

**Claims Explanation:**

I have a couple of comments about some of the claims made in the paper.

*On the augmentation scheme:*

1- I do not really understand the motivation: "Our key insight is that meaningful graph views should effectively maximize spectral differences while preserving critical structural properties by introducing controlled variations based on centrality-guided augmentations."  Could the authors provide a little bit more intuition or justification on this? Why centrality? Why not preserve motifs of interest?

2- If I understand correctly, there is no feature modification on the first graph --- is that a limitation? I find the explanation of the adversarial bootstrapping extremely confusing. Am I right in understanding that $\mathcal{G}_1$ has similar laplacian  but different features than the original graph, and the opposite for $\mathcal{G}_2$?

3-  If I understand correctly, one of the pitches of this paper is that there are no custom augmentation schemes (e..g hyperparameters  to choose to appropriately augment the graph). However, the authors introduce a budget constraint $B$ and a learning rate $\eta$, which look like they could be crucial... More specifically, in my understanding, the learning rate $\eta$, if high, will push the graph $\mathcal{G}_1$ to be far away from the original graph. If small, the perturbation will be tiny. The budget $B$ will have the opposite effect. The authors do not discuss the effect of these parameters in any point of the paper. This severely undermines the potential contribution made by this paper.

4- The modifications introduced on the graph structure look a little strange themselves. More specifically, I follow the authors' argument p 18: "The optimization objective is defined through the spectral distance: [...]". I do not follow however how this loss is indeed what the authors optimize: "Specifically, L(∆) corresponds to the max and min updates for the augmentation matrices and is therefore embedded within the Laplace optimization described in Appendix A.2.1".Looks to me like the gradient for their loss would be   $L(\Delta) = \frac{1}{\| \lambda(tilde{L}) - \lambda(\tilde{L}_{mod}}\|} \times ( 2/ \lambda(\tilde{L}) + 2 \lambda(\tilde{L}_{mod})/ \lambda(\tilde{L})^2 })$.

So I don't completely follow the logic behind the proposed updates. To me, it does not agree with the initial goal. I also find strange that $\mathcal{G}_1$ and $\mathcal{G}_2$ are completely different (one maximizes the spectral divergence the other minimizes it). To the best of my knowledge, in self supervised learning, the graphs generated are usually assumed to be i.i.d copies.  I can appreciate if the authors made a different choice, but highlighting this with a bit of discussion could be enlightening.

5- The authors should report visualization and statistics on the graphs that are created. It would be good to report summary statistics on the types of graph created (number of edges, centrality, etc) as well as visualizations --- which need to be contrasted to other graph augmentation schemes.

6- The sentence: "This general formulation allows our method to flexibly adapt to diverse graph types and tasks." seems misleading. The method is completely task independent --- the weights are solely dependent on the graph structure.

7-  The ablation studies are not complete. To disentangle the benefits of the augmentation from the adversarial bootstrapping, the authors would need to test how their augmentation schemes would perform deployed on other architectures (e.g. MVGRL, GRACE), and reciprocally, how their adversarial bootstrapping would fare using another data augmentation scheme.  I would also report the value of the coefficient alpha for the different datasets --- are certain centrality scores better for certain graphs, or do all datasets yield similar choices of centrality weights?

8 - The authors report not requiring negative pairs, but the complexity of the algorithm is still O(n^2) --- at least in the preprocessing. The time complexities reported in practice in table 7 seem fine though --- do they include the preprocessing time? Is that the time for the entire method, from the graph augmentation to the adversarial bootstrapping? Another main comment here is that I would suggest trying out as well CCA-SSG, another self-supervised learning approach that does not require negative pairs and that is generally quite fast and works pretty well.



*On the robustness to adversarial attacks:*

1- One of the authors' claims is that usual self-supervised methods can be quite sensitive to adversarial graph perturbations. However, the reference for this claim is a paper that studies sensitivity in the context of classification.The fact that the same phenomenon holds for self supervised methods is debatable. Intuitively, a lot of the self-supervised methods create alternative versions of the graph by dropping edges, masking features --- eg. the same attacks that can be done to destabilize the GNN. Consequently, one could expect the attacks to not affect the self supervised methods as much as for the supervised setting.

2- It would be great to detail in the main text the difference between the four types of attacks that are performed.

3- It is unclear to me how the results for the benchmark methods have been obtained for this section. To the best of my knowledge, the benchmarks rely on several hyperparameters (feature mask rate, edge drop rate) that are tunable. Have the authors optimized these parameters in the table? Or are they using the default parameters? To be clear, I completely agree with the authors that these hyperparameters are extremely obnoxious to tune, but one could argue that they need to be tuned to allow a fair comparison.

**Requested Changes:**

Minor (Typos):
1- The sentence "More recently, masked autoencoder approaches like GraphMAE (Hou et al., 2022) and AugMAE (Wang et al., 2024) have demonstrated strong performance by reconstructing masked graph components" is repeated twice (p2).

2- There is an inconsistency in the updates p4 and described in the algorithm (l18 and 20, the $\|\lambda_{mod1}\|^2$  vs $\|\lambda_{mod1}\|$ p4).

3- Algorithm 1 is a little strange. The first part describes the augmentation of one laplacian. But line 23 mentions "for each graph G in the dataset". I assume a different laplacian is generated for each graph though, so that $\Delta_1$ and $\Delta_2$ are different for each graph?

---

### Decision · Action_Editor_tp5G · 2026-01-09

**Recommendation:** Reject

**Audience:**

Yes

**Audience Explanation:**

Although one of the reviewers questions the novelty and expresses concerns that the contribution is marginal, other reviewers acknowledge that the paper proposes a method that is likely to be of interest to TMLR readers working on graph learning (provided the methodology is correct and clearly explained).

**Claims And Evidence:**

No

**Claims Explanation:**

As indicated by one of the reviewers, even after revisions, there remain mismatches between the material in the response to the reviews, the main body of the paper, and the appendices. There are notational inconsistencies that make the proposed approach difficult to understand and verify. Indeed, in the final response the authors write "We are committed to ensuring precise mathematical consistency in the final version.", which implies that it is not yet there and there is a need for revision and another review.

**Resubmission Of Major Revision:**

The authors may consider submitting a major revision at a later time.